# Redox-Triggered Switching of Conformational State in Triple-Decker Lanthanide Phthalocyaninates

**DOI:** 10.3390/molecules27196498

**Published:** 2022-10-01

**Authors:** Alexander G. Martynov, Marina A. Polovkova, Yulia G. Gorbunova, Aslan Yu. Tsivadze

**Affiliations:** 1Frumkin Institute of Physical Chemistry and Electrochemistry, Russian Academy of Sciences, Leninsky pr., 31, Bldg. 4, 119071 Moscow, Russia; 2Kurnakov Institute of General and Inorganic Chemistry, Russian Academy of Sciences, Leninsky pr., 31, 119991 Moscow, Russia

**Keywords:** triple-decker phthalocyaninates, conformation, UV-vis-NIR spectroscopy, simplified Tamm–Dancoff approximation, noncovalent interactions, reduced density gradient, quantum theory of atoms in molecules

## Abstract

Double- and triple-decker lanthanide phthalocyaninates exhibit unique physical-chemical properties, particularly single-molecule magnetism. Among other factors, the magnetic properties of these sandwiches depend on their conformational state, which is determined via the skew angle of the phthalocyanine ligands. Thus, in the present work we report the comprehensive conformational study of substituted terbium(III) and yttrium(III) trisphthalocyaninates in solution depending on the substituents at the periphery of molecules, redox-states and nature of solvents. Conjunction of UV-vis-NIR spectroscopy and quantum-chemical calculations within simplified time-dependent DFT in Tamm–Dancoff approximation provided the spectroscopic signatures of staggered and gauche conformations of trisphthalocyaninates. Altogether, it allowed us to demonstrate that the butoxy-substituted complex behaves as a molecular switcher with controllable conformational state, while the crown-substituted triple-decker complex maintains a staggered conformation regardless of external factors. The analysis of noncovalent interactions within the reduced density gradient approach allowed to shed light on the nature of factors stabilizing certain conformers.

## 1. Introduction

Sandwich double- and triple-decker lanthanide (Ln) complexes with phthalocyanine (Pc) ligands possess numerous useful properties [1,2,3]. Among them, the single molecule magnetism of Tb(III), Dy(III) and Er(III) complexes is particularly attractive [4,5], rendering these complexes as valuable building blocks in molecular memory devices and spintronic technologies [6]. Moreover, the sensitivity of magnetic properties to external stimuli affords their application as molecular switches, i.e., components of smart materials [7].

The characteristic structural features of sandwich complexes which have particularly pronounced effects on their magnetic properties are the interligand distance *d* and skew angles *θ* between stacked ligands (Figure 1a–c). The *d* value can be described as a separation between N_4_ centroids of tetrapyrrolic ligands, and it dictates the ligand field strength which in turn determines the height of magnetization relaxation barrier [8]. The *θ* value defies the symmetry of lanthanide ion coordination surrounding and affects the magnetization relaxation mechanisms [9,10]. While the distance *d* is clearly governed mainly by the size of the bridging metal center [11] and the overall redox-state of the sandwich complex [12], the skew angle *θ* reveals trickier dependence on structural [13], electronic [14] and supramolecular factors [15].

One example of such intricate behaviour relates to the widely studied triple-decker Ln(III) phthalocyaninates with alkoxy- and crown-ether substituents [16]. In the following paper, we will discuss this behaviour on the examples of octa-*n*-butoxy- and tetra-15-crown-5-substituted Pc ligands (Figure 1d).

The first synthesis and characterization of triple-decker complexes **M_2_[(BuO)_8_Pc]_3_**, based on octa-*n*-butoxy-phthalocyanine was reported by Takahashi et al. in 1993 on the examples of sandwiches formed by Dy(III), Yb(III), La(III), and Lu(III) complexes [17,18]. The authors noted the dramatically different appearance of the UV-vis spectrum of the triple-decker complexes in dichloromethane (broad Q-band with several unresolved inflexions) and benzene (sharp Q-band with well-resolved Q_1_ and Q_2_ components).

Moreover, similar solvatochromism was observed on the examples of various classes of alkoxy-substituted Pc sandwiches, including siloxane dimers **RO[(AkO)_8_PcSiO]_2_R** [19,20] and bisphthalocyaninates with µ-nitribo-diiron and µ-carbido-diruthenium cores **[(AlkO)_8_PcM]_2_(µ-X)**, M = Fe, X = N [21] and M = Ru, X = C [22]. Detailed NMR characterization of siloxane bisphthalocyaninates allowed to explained the difference in spectral properties in terms of conformers with different skew angles [20]. Thus, the gauche (***g****-*) form with *D*_4_ symmetry and the skew angle *θ* ≈ 30° was stabilized in chloroalkanes. The staggered (***s****-*) form with *D*_4*h*_ symmetry and the angle of *θ* ≈ 45° was stabilized in aromatic solvents. The preferable formation of one or another conformer was only tentatively explained by the difference in attractive interactions between solvent molecules and aromatic systems of Pc ligands. TD-DFT calculations of siloxane dimers allowed to differentiate their ***s****-* and ***g***-conformers [23,24].

The gauche form was always found by XRD in all crystals of neutral **M_2_[(BuO)_8_Pc]_3_**, M = Gd [25], Tb [26], Dy [27]. However, in 2020, Horii et al. described the crystal structure of the dicationic complex **{Tb_2_[(BuO)_8_Pc]_3_}^2+^(SbCl_6_^−^)_2_**, where two electrons were removed from π-orbitals of Pc ligands. In contrast to the parent neutral complex, triple-decker dication adapted the staggered conformation [14]. Importantly, oxidation caused decrease of interligand distance by 0.067 Å, which resulted in decrease in the magnetic anisotropy as evidenced from both theoretical calculations and *ac* magnetic susceptibility measurements. Altogether, these results suggest that the conformational state of alkoxy-substituted triple-deckers can be influenced either by solvation or by redox-transition.

Different conformational behaviour was observed in the case of crown-substituted triple-deckers. Although their UV-vis spectra in aromatic solvents were not studied, the spectra in chloroalkanes were typical for ***s***-conformer [22,28,29,30,31,32] in clear contradiction with spectra of BuO-substituted counterparts. Moreover, X-ray diffractometric studies of crown-trisphthalocyaninates single crystals evidenced that pairs of **[(15C5)_4_Pc]** ligands always adapt staggered conformations [31,32,33,34]. Only in the case of the heteroleptic complexes **[(15C5)_4_Pc]M*[(15C5)_4_Pc]M(Pc)** could the coordination polyhedron of the M* metal centre be switched from square-antiprismatic to square-prismatic via intercalation of potassium cations between crown-substituted ligands forcing the formation of the eclipsed conformation [15].

Thus, in the present work, we aimed at a comparative study of the conformational behaviour of alkoxy- and crown-substituted Tb(III) and Y(III) trisphthalocyaninates depending both on the solvent and oxidation degree of the complexes. Time-dependent DFT calculations in simplified Tamm–Dancoff approximation (sTDA) were used to rationalize the observed spectral behaviour.

Although the chosen metal centres have essentially different electronic nature (Tb([Xe]f^9^s^2^) and Y([Kr]d^1^s^2^), the synthesized complexes exhibit almost identical optical properties. Thus, Y(III) complexes were used as references for quantum-chemical calculations which could be reliably extrapolated to Tb(III) complexes having advantageous magnetic properties. Altogether, it provided a comprehensive summary of factors affecting the spectral properties of substituted trisphthalocyaninates, which revealed spectral signatures indicative of the conformational states of the complexes in solution.

## 2. Results and Discussion

### 2.1. Synthesis of Trisphthalocyaninates

Although the synthesis of Tb(III) and Y(III) trisphthalocyaninates with BuO- and 15C5-substituted ligands is already documented, we were able to improve synthetic protocols, which allowed faster procedures and higher yields. Previously, 1-octanol or 1-chlofornaphthalene were used as solvents which were used for the interaction between **H_2_[(BuO)_8_Pc]** or **H_2_[(15C5)_4_Pc]** with corresponding acetylacetonates affording target triple-deckers in ~20% [26] and ~50% [35] after 4 h and 1.5 h of reaction mixture refluxing respectively. Herein, we found that application of a 9:1 vol. mixture of 1,2,4-trichlorobenzene and 1-octanol as a solvent allowed to increase the yields of **Tb_2_[(BuO)_8_Pc]_3_** and **Tb_2_[(15C5)_4_Pc]_3_** to 90% and 68% respectively. The yields for Y(III) counterparts were 75% and 79% respectively. Importantly, the reaction times were reduced to 30 min. Analytical characteristics of thus synthesized complexes were in agreement with the previously reported data (Appendix A).

### 2.2. Solvatochromic Behaviour of Trisphthalocyaninates Depending on the Nature of Substituents and Redox State

UV-vis spectra of the synthesized Tb(III) and Y(III) trisphthalocyaninates were measured in dichloromethane and benzene (Figure 2). The pronounced difference in the spectral appearance of **M_2_[(BuO)_8_Pc]** in aliphatic and aromatic solvents is in line with the previous report [17,18]. However, in the case of **M_2_[(15C5)_4_Pc]_3_**, only a negligible difference can be noted. To the best of our knowledge, this feature of crown-substituted complexes has never been reported before, and this result can be used as a marker of invariance of conformational state of the latter complex on either the aliphatic or aromatic nature of solvent.

The subtle difference in Q-bands wavelengths of Tb(III) and Y(III) complexes is explained by small variations in ionic radii of metal centres which govern the interligand distance and intramolecular interactions between stacked ligands [36,37].

Further study of oxidation of complexes in CH_2_Cl_2_ or C_6_H_6_ was performed using phenoxathiinylium hexachloroantimonate, **OxSbCl_6_** as a mild oxidant which has already been used for the stepwise oxidation of tetrapyrrolic sandwiches [14,38,39,40].

Figure 3 show the UV-vis-NIR spectra of the neutral, one- and two-electron oxidized forms of the complexes **Tb_2_[(BuO)_8_Pc]_3_** and **Tb_2_[(15C5)_4_Pc]_3_**, analogous spectra for Y(III) complexes are shown in Appendix A. Whereas in the spectra of neutral trisphthalocyanates there is no absorption in NIR region, the presence of absorption band at ca. 2300–2500 nm is characteristic for monocations of these complexes (accurate determination of the maximum of this band is hampered by the overlap with the solvent absorption bands). The second oxidation leads to a hypsochromic shift of the NIR band to 1700–1800 nm. The trend in the shift of the Q-band upon oxidation is opposite—removal of one and two electrons is followed by bathochromic shift of this band. Oxidation of triple-decker complexes is also followed by the appearance of new bands at 450–530 nm. While the near-IR region has already been recognized as the informative region for the identification of the redox states of sandwich complexes [36,41,42], we found that the bands in this region can be also used to identify the conformational state of complexes.

Unfortunately, the oxidized forms of the crown-substituted complex rapidly precipitated from benzene solution, therefore their high-quality spectra could not be acquired, however the aforementioned assumption of the conformational rigidity of this complex allowed us to use the spectral patterns of its redox-forms in CH_2_Cl_2_ as markers for staggered conformations. Thus, we compared the UV-vis-NIR spectra of the 1*e*-oxidized complexes **Tb_2_[(BuO)_8_Pc]_3_^+^** and **Tb_2_[(15C5)_4_Pc]_3_^+^** in CH_2_Cl_2_, where parent neutral complexes exist in gauche and staggered conformations respectively. In addition to the notable difference in the shape of the Q-bands of the complexes, the striking difference in the shape and number of absorption bands in the NIR region must also be noted—while in the spectrum of the crown-substituted complex one wide band with a maximum at 2450 nm is observed, in the spectrum of butoxy-substituted complex contains additional weak bands at 1250–1750 nm. Moreover, the UV-vis-NIR spectrum of **Tb_2_[(BuO)_8_Pc]_3_^+^** in benzene shows only one broad band at 2650 nm. Assuming that benzene stabilizes staggered conformations, we can attribute the bands at 1250–1750 nm to the marker of the gauche conformation.

To the contrast, the shape and appearance of spectra of dicationic forms of both complexes in CH_2_Cl_2_ are nearly identical, suggesting the uniformity of their conformational state, and it is reasonable to ascribe the observed spectral pattern to the staggered conformation based on crystallographic characterization of **{Tb_2_[(BuO)_8_Pc]_3_}^2+^(SbCl_6_^−^)_2_** [14].

### 2.3. Quantum-Chemical Modelling of Conformation- and Redox-Dependent UV-vis-NIR Spectra of Trisphthalocyaninates

Altogether spectral observations suggest that 1*e*-oxidation of **M_2_[(BuO)_8_Pc]_3_** in CH_2_Cl_2_ *does not* cause switching of its conformation from gauche to staggered, however 2*e*-oxidation *does* result in such switching. The complex **M_2_[(15C5)_4_Pc]_3_** is assumed to be conformationally invariant. To support these conclusions, we performed quantum-chemical calculations to predict UV-vis-NIR spectra of redox forms of triple-deckers depending on their conformational state. With this aim, we optimized **s**- and ***g***-conformers of neutral, cationic and dicationic forms of Y(III) trisphthalocyaninates where butoxy- and crown-substituents were truncated and replaced with MeO-groups.

Optimization was performed in ORCA 5.0.3 package [43] using BP86 DFT functional, def2-SVP basis set for light atoms and def2-ECP for yttrium. Although this computational level can be considered as a relatively modest, it was previously shown that it provides sufficiently accurate geometries for prediction of spectral properties of phthalocyanines and related compounds [15,44,45] which is particularly attractive from the viewpoint of the computational cost of large molecules.

Importantly, Grimme’s atom-pairwise dispersion correction and Becke–Johnson damping (D3BJ) was used to reproduce the intramolecular interactions between stacked Pc ligands in sandwich complexes [46]. In the absence of this correction, geometrical optimization of sandwich phthalocyaninates lead to structures with severely concaved Pc ligands which clearly contradicts X-ray data [47]. On the other hand, it has already been reported that the applied D3BJ correction overestimates the stabilisation of the ***g***-conformations of alkoxy-substituted sandwiches even in cases where such conformation can be excluded on the basis of experimental data [48].

Indeed, in our case neutral molecules and both oxidized forms converged to ***g***-conformers, so the ***s***-geometries could be obtained only by imposing the geometrical constrains setting the skew angles to 45°. BP86/def2-SVP gas-phase energies of the converged geometries were always larger for the gauche forms for each redox-state, although the difference ∆E between energies of gauche and staggered conformers systematically decreased with the increase of the molecular charge. Accounting for solvation with benzene and dichloromethane using the implicit SMD model [49] neither changes the geometry nor affected the relative stability of certain forms.

For more accurate evaluation of relative stabilities of conformers, we performed single-point calculations for the converged geometries using *r*^2^SCAN-3c [50], a “Swiss army knife” composite electronic-structure method, which shows a spectacular performance and robustness for reaction and conformational energies as well as non-covalent interactions. A comparison of energies, calculated for gas-phase and implicit SMD surrounding, suggests gradual destabilization of ***g***-conformations upon stepwise oxidation making ***s***-conformation the most stable in all media, making the ***s***-dication the most stable form in both solvents, which correlates well with the aforementioned spectroscopic data (Table 1). However, even the advanced *r*^2^SCAN-3c method cannot reproduce other features, including the existence of neutral and cationic alkoxy-substituted trisphthalocyaninates in ***s***-forms in aromatic media and ***g***-forms in chloroalkanes. As there is no explicit solvation in our calculations, we can make the cautious assumption that these results can evidence of the role of specific solvation due to weak solvent/solvate interactions, which may involve hydrogen bonds and π-π stacking.

Analysis of structural characteristics of the optimized geometries (Figure 4) evidence that oxidation results in gradual contraction of Y…Y and N_4_…N_4_ distances (Table 2). The fair agreement of computational results with crystallographic data for reported triple-decker complexes justifies the adequacy of the selected structural model. The noteworthy difference in structures of **s**- and ***g***-conformers is the systematic contraction of the metal-metal distance in the latter case which may have impact on *f*-*f* interactions in trisphthalocyaninates bearing two paramagnetic lanthanide centres. This impact is yet to be studied experimentally.

Optimized structures were used to predict the energies of vertical excitations in UV-vis-NIR spectra of trisphthalocyaninates within the simplified Tamm–Dancoff approximation (sTDA) [51,52,53] with CAM-B3LYP functional [54], def2-SVP basis set for light atoms and def2-ECP for yttrium. Previously, both simplified TDA and TDDFT were demonstrated to afford spectacular orders of magnitude speedup of calculations in comparison with full TDDFT without loss of accuracy or even providing more accurate results for prediction of UV-vis-NIR spectra of huge molecules, and they were widely applied for phthalocyanines and related compounds [45,55,56,57]. Herein, we used this method for the first time to treat triple-decker complexes, thus contributing to the history of successful use of simplified time-dependent DFT approximations.

Frontier molecular orbitals responsible for the appearance of bands in UV-vis-NIR spectra of trisphthalocyaninates are formed from linear combinations of Pc-centred orbitals, i.e., HOMOs and pairs of degenerate LUMOs [36,58,59]. Depending on the molecular symmetries, the contributions of parent ligand orbitals can form bonding, nonbonding, and antibonding combinations (Figure 5). Thus, both conformations have the bonding and antibonding nature of HOMO-2 and HOMO respectively. However, HOMO-1 of ***g***-conformer also becomes a bonding orbital with a significant contribution from the inner ligand, while the contribution from this ligand in ***s***-form is zero. The antibonding nature of the neutral triple-decker HOMO is responsible for the decrease of the interligand distance upon the stepwise removal of electrons from this orbital.

Excitation from the three highest occupied MOs to three pairs of LUMOs is responsible for Q-bands observed in the visible range. Sequential oxidation of the complex leads to new vacant orbitals, the electronic transitions to which give rise to bands in the near-infrared region. Although these trends are common for both conformations, the comparison of diagrams of frontier MOs evidence that altering the molecular symmetry strongly affects the energies occupied orbitals (Figure 6), which inevitably has a profound effect on energies and configurations of vertical excitations (Appendix A).

A comparison of the spectra, calculated for the ***s***-conformers of neutral and oxidized forms of **Y_2_[(MeO)_8_Pc]_3_** and experimental spectra of redox-forms of **Y_2_[(15C5)_4_Pc]_3_** in CH_2_Cl_2_, evidences their excellent agreement (Figure 7). Thus, BP86 structural model with further CAM-B3LYP sTDA treatment is a fortunate computational combination which accurately reproduces spectral features of trisphthalocyaninates, including the Q-bands bathochromic shift upon oxidation, the appearance of the NIR band in 1*e*-oxidized complex, and the hypsochromic shift of NIR band upon 2*e*-oxidation. The errors in positions of Q-band NIR bands are in the ranges of 0.08–0.18 eV and 0.03–0.04 eV respectively, which does not exceed the typical errors in TD-DFT calculations of phthalocyanines and related compounds [54]. Moreover, the analogy between spectral appearances of complexes with different metal centres allows us to extrapolate the results obtained for the Y(III) complexes to Tb(III) counterparts.

The excellent agreement between simulated and measured spectra validates sTDA model and justifies it for further prediction of the experimentally unobtainable spectrum of the dicationic gauche complex. Modelling suggests that this spectrum should have dramatically different appearance in NIR range—the band at 3500 nm is expected instead of absorbance at 1800 nm (Figure 8). Interestingly, such exceptionally low-lying π-π* excited states of sandwich phthalocyaninates were observed previously by Fukuda and Ishikawa et al. in spectra of oxidized forms of quadruple-decker phthalocyanine complexes [39]. However, in that case, the appearance of this band was not related to conformational effects but was ascribed to the consolidated conjugation through the huge molecules composed of four stacked ligands.

### 2.4. Analysis of Noncovalent Interactions Stabilizing Conformations of Trisphthalocyaninates

With the CAM-B3LYP/def2-SVP wave functions of the trisphthalocyaninates in hand, we tried to identify factors that stabilise their conformations. For this purpose, we performed graphical visualisation of noncovalent interaction (NCI) isosurfaces based on reduced gradient density (RDG, Equation (1)) [60].
(1)$$RDG\left(r\right)=\frac{1}{2{\left(3\pi^{2}\right)}^{2/3}}\cdot \frac{\left[∆\rho \left(r\right)\right]}{{\left[\rho \left(r\right)\right]}^{4/3}}$$

Here *ρ* is electron density and **r** is coordinate vector. Isosurface map of RDG at low electron density area gives illustrative image of noncovalent interactions where areas of strong attraction and repulsion are typically coloured into blue and red, green colour corresponds to weaker Van der Waals interactions (Figure 9). According to the NCI-RDG method, the strength and type of interaction can be identified by *sign*(*λ*_2_)*ρ*, where *sign*(*λ*_2_) is the sign of the second largest eigenvalue of electron density Hessian matrix at position **r**. Thus, *sign*(*λ*_2_)*ρ* will be positive for repulsive interactions and negative for attractive interactions. Plotting the RDG value vs. *sign*(*λ*_2_)*ρ* gives scattered plots where spikes with different signs and magnitudes correspond to noncovalent interactions ranging from H-bonding to van der Waals interactions and steric repulsion.

RDG isosurfaces plotted for **Y_2_[(MeO)_8_Pc]_3_** clearly show that the area of attraction between stacked ligands is wider for the ***g***-conformer which correlates with stronger overlap between these ligands. Moreover, this region spreads to the peripheral substituents of the molecule, whereas in the case of the ***s***-conformer it breaks off without reaching the periphery.

Complementary calculations within quantum theory of atoms in molecules (QTAIM) were also performed to find the critical points corresponding to certain noncovalent interactions [61]. They suggest that methoxy-groups are involved in stabilization of the gauche form as there are numerous (3,−1) critical points corresponding to CH…O contacts, and these points lie on the attractive regions NCI isosurfaces. To the contrast, in the case of the staggered form, only weak H…H interactions can be identified according to QTAIM.

RDG vs. *sign*(*λ*_2_)*ρ* plots for two conformations of neutral **Y_2_[(MeO)_8_Pc]_3_** (Figure 10) show that contributions from attractive interactions in ***g***-conformer are larger than in ***s***-forms, and an additional spike is observed in the near-zero region of the formed plot. Analogous plots drawn for other redox forms (ESI) reproduce the same trends but they are not sufficient for the definitive identification of interactions which cause the switching of conformers.

Altogether, these results again confirm that the effects responsible for the stabilization of certain conformers are relatively subtle. Thus, the applied truncated models which do not include the explicit interaction of substituents in trisphthalocyaninates with solvent molecules can only be cautiously used for the description of the conformational behaviour of these sandwich complexes. However, in conjunction with spectroscopic data, these models can provide reliable information about the conformational state of complexes in solution.

## 3. Materials and Methods

### 3.1. Physical-Chemical Measurements

All solvents were distilled over appropriate drying agents prior to use. All starting reagents were applied from commercial suppliers. Phthalocyanines **H_2_[(BuO)_8_Pc]** and **H_2_[(15C5)_4_Pc]** [62] and the oxidizing agent phenoxathiinylium hexachloroantimonate, **OxSbCl_6_** [38], were synthesized according to the previously reported procedures.

UV–vis–NIR spectra in the range of 250–3000 nm were measured using a spectrophotometer V-770 (JASCO) in quartz cells with 1 cm optical path. Matrix-assisted laser desorption ionization time-of-flight (MALDI-TOF) mass spectra were measured on a Bruker Daltonics Ultraflex spectrometer. Mass spectra were registered in positive ion mode using 2,5- dihydroxybenzoic acid as a matrix. MALDI-TOF mass spectra of synthesized complexes are provided in Appendix A. NMR spectra were recorded using a Bruker Avance III spectrometer with 600 MHz proton frequency in CDCl3 at ambient temperature with the use of the residual solvent resonance as internal reference. NMR spectra of synthesized complexes are provided in Appendix A. Analytical characteristics of the synthesized complexes were in agreement with the previously reported data [63,64].

### 3.2. Synthesis and Characterization of Phthalocyanines

***Diterbium(III) tris(octa-butoxy-phthalocyaninate)*****Tb_2_[(BuO)_8_Pc]_3_.** Terbium acetylacetonate (140 mg, 0.29 mmol) was added to a boiling solution of octa-butoxy phthalocyanine **H_2_[(BuO)_8_Pc]** (107 mg, 98 µmol) in a mixture of 4.5 mL of 1,2,4-trichlorobenzene (TClB) and 0.5 mL of 1-octanol in an argon atmosphere and the reaction mixture was refluxed for 30 min. After cooling, the reaction mass was transferred to a chromatographic column filled with aluminium oxide in a mixture of chloroform/hexane 3:2 vol. Elution with the same mixture allowed separation of trichlorobenzene. Further, a mixture of chloroform/hexane 4:1 vol. was used to isolate the target triple-decker complex in mixture with octanol. After evaporation of volatile solvents, 25 mL of methanol was added to the obtained oily mixture. The resulting suspension was kept in an ultrasonic bath for 30 min, and the target complex was filtered off and dried at 80 °C overnight, affording **Tb_2_[(BuO)_8_Pc]_3_** as a dark blue fine-crystalline powder (106 mg, 90%). MALDI TOF: *m*/*z* calcd for C_192_H_240_N_24_O_24_Tb_2_ 3585.7, found 3585.9 [M^+^]. UV–Vis–NIR (benzene) λ_max_ (nm) (log ε): 293 (5.18), 362 (5.29), 644 (5.52), 692 (4.70). UV–Vis–NIR (CH_2_Cl_2_) λ_max_ (nm) (log ε): 293 (5.17), 352 (5.20), 554 (4.52), 643 (5.04).

***Diyttrium (III) tris(octa-butoxy-phthalocyaninate)*****Y_2_[(BuO)_8_Pc]_3_.** The complex (32 mg, 75%) was synthesized using the aforementioned procedure starting from octa-butoxy-phthalocyanine **H_2_[(BuO)_8_Pc]** (40 mg, 37 µmol) and yttrium acetylacetonate (42 mg, 0.11 mmol). MALDI TOF: *m*/*z* calcd for C_192_H_240_N_24_O_24_Y_2_ 3445.6, found 3445.4 [M^+^]. UV–Vis–NIR (benzene) λ_max_ (nm) (log ε): 293 (5.15), 361 (5.28), 643 (5.44), 698 (4.70). UV–Vis–NIR (CH2Cl2) λ_max_ (nm) (log ε): 293 (5.17), 352 (5.22), 547 (4.52), 643 (5.13). 

***Diterbium(III) tris(tetra-15-crown-5-phthalocyaninate)*****Tb_2_[(15C5)_4_Pc]*_3_***. Terbium acetylacetonate (112 mg, 2.46 mmol) was added to a boiling solution of tetra-15-crown-5-phthalocyanine (104 mg, 0.82 mmol) in a mixture of 4.5 mL of 1,2,4-trichlorobenzene (TClB) and 0.5 mL of 1-octanol in an argon atmosphere and the reaction mixture was refluxed for 30 min. After cooling, the reaction mass was transferred to a chromatographic column filled with aluminium oxide in a mixture of chloroform/hexane 3:2 vol. Elution with the same mixture allowed the separation of trichlorobenzene followed by octanol. Further, a mixture of chloroform + 1.75 vol.% MeOH was used to isolate the target triple-decker complex. After evaporation of volatile solvents, the residue was dried at 80 °C overnight affording **Tb_2_[(15C5)_4_Pc]_3_** as a dark blue powder (72 mg, 68%). MALDI TOF: *m*/*z* calcd for C_192_H_216_N_24_O_60_Tb_2_ 4137.3, found 4138.1 [M + H]^+^. UV–Vis–NIR (benzene) λ_max_ (nm) (log ε): 292 (5.14), 362 (5.25), 645 (5.45), 693 (4.66). UV–Vis–NIR (CH_2_Cl_2_) λ_max_ (nm) (log ε): 292 (5.02), 363 (5.11), 644 (5.22), 698 (4.61).

***Diyttrium(III) tris(tetra-15-crown-5-phthalocyaninate)*****Y_2_[(15C5)_4_Pc]*_3_***. The complex (42 mg, 79%) was synthesized using the aforementioned procedure starting from tetra-15-crown-5-phthalocyanine **H_2_[(15C5)_4_Pc]** (51 mg, 40 µmol) and yttrium acetylacetonate (48 mg, 0.12 mmol). MALDI TOF: *m*/*z* calcd for C_192_H_216_N_24_O_60_Y_2_ 3997.3, found 3997.2 [M]^+^. UV–Vis–NIR (benzene) λ_max_ (nm) (log ε): 293 (5.16), 362 (5.28), 644 (5.43), 699 (4.71). UV–Vis–NIR (CH2Cl2) λ_max_ (nm) (log ε): 293 (5.12), 363 (5.22), 642 (5.32), 705 (4.68).

### 3.3. Spectrophotometric Investigation of Trisphjthalocyaninates Oxidation

Aliquots of solutions of complexes in CH_2_Cl_2_ or C_6_H_6_ (*ca.* 10^−5^ M) were placed into quarts cells with Teflon stoppers, and solution of phenoxathiinylium hexachloroantimonate, **OxSbCl_6_** in CH_2_Cl_2_ (3.7 mM) was added stepwise in 5 µL portions. UV-Vis-NIR spectra were measured after each addition in the range of 250–3000 nm. Each oxidation was characterized by its own set of isosbestic points evidencing the stepwise conversion of neutral forms to cations followed by the oxidation of cations to dications.

### 3.4. Quantum-Chemical Modelling

All calculations were performed using ORCA 5.0.3 quantum chemical package [43] for yttrium complexes where all peripheral substituents were truncated to methoxy-groups. Thus, the structures of the resulting model complexes **Y_2_[(MeO)_8_Pc]_3_** in neutral, mono-, and dicationic forms were optimized using BP86 functional and def2-SVP basis set [65]. Geometrical constraints were imposed to obtain geometries with staggered conformations, gauche conformations were optimized without any constraints.

The geometry optimization was performed using Grimme’s atom-pairwise dispersion correction and Becke–Johnson damping (D3BJ) [66]. The energies within the range of 0–5 eV and oscillator strengths of vertical excitations were calculated using simplified Tamm–Dancoff approximation [45,51,52,53] with CAM-B3LYP functional and def2-SVP basis set. Optimization and sTDA calculations were performed in gas phase, as well as dichloromethane or benzene media. Solvation effects were accounted for by using the solvation model based on density (SMD) [49]. However, both gas-phase and SMD calculations showed that accounting for solvation has a negligible effect on geometries, excitation energies and oscillator forces.

Gabedit 2.3.0 program was used to prepare the input files and to follow the progress of the calculations [67], and the Chemissian 4.65 program (by L. Skripnikov) was used to analyse and visualize the results of the quantum chemical calculations. For the current version, see www.chemissian.com, accessed on 1 September 2022.

Reduced density gradient (RDG) analysis and calculations within Quantum theory of atoms in molecules (QTAIM) were performed using Multiwfn 3.8 (dev) [68] and VMD 1.9.4 [69] was used for visualization. Calculations were performed using CAM-B3LYP/def2-SVP wavefunctions.

## 4. Conclusions

The main outcome of our work is the establishment and interpretation of spectroscopic signatures that can be used to determine the conformational states of trisphthalocyaninates in solution in visible and near-IR ranges. The influence of skew angles of phthalocyanine ligands in such complexes on their physical-chemical properties, including single-molecule magnetism [9,10] or nonlinear optical behaviour [70], justifies the value of these correlations as solutions are commonly used to produce Pc-bases materials. On the other hand, our work highlights the need to critically evaluate the results of quantum chemical calculations of nonrigid molecules, where combining spectroscopic data and appropriate theoretical models is particularly important.

## Figures and Tables

**Figure 1 molecules-27-06498-f001:**
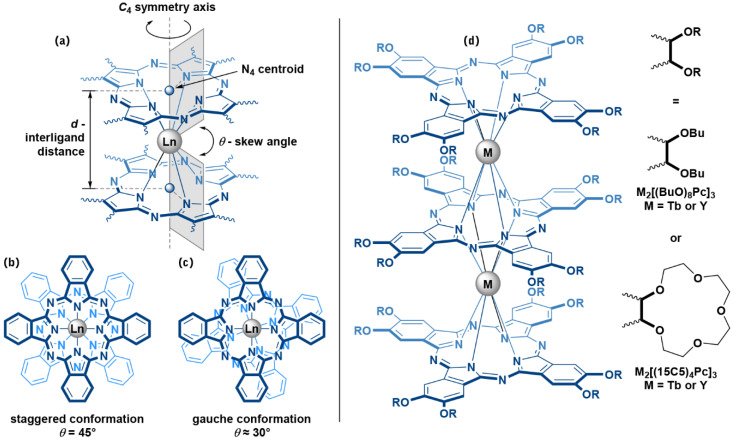
(**a**) Structural characteristics *d* and *θ*, which determine coordination surrounding of lanthanide ion sandwiched between two Pc ligands; (**b**,**c**)—staggered (***s****-*) and gauche (***g****-*) conformations of Pc ligands in sandwich complexes; (**d**)—trisphthalocyaninates studied in the present work—**M_2_[(BuO)_8_Pc]_3_**—diterbium(III) or diyttrium(III) tris(octa-*n*-butoxyphthalocyaninates) and **M_2_[(15C5)_4_Pc]_3_**—diterbium(III) or diyttrium(III) tris(tetra-15-crown-5-phthalocyaninates).

**Figure 2 molecules-27-06498-f002:**
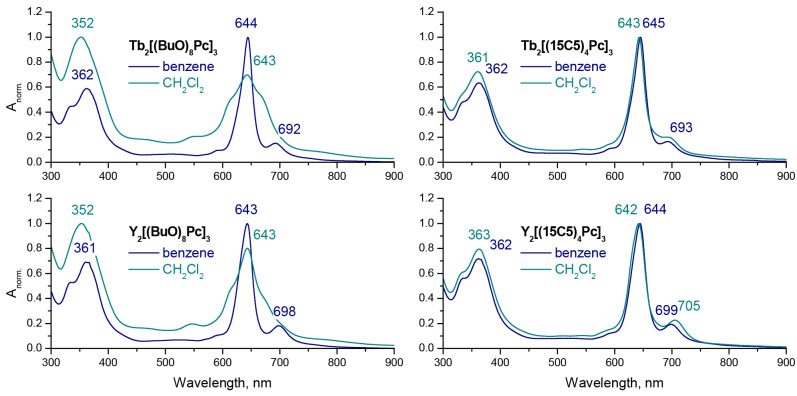
Normalized UV-vis spectra of **M_2_[(BuO)_8_Pc]_3_** and **M_2_[(15C5)_4_Pc]_3_**, M = Tb and Y in benzene and dichloromethane.

**Figure 3 molecules-27-06498-f003:**
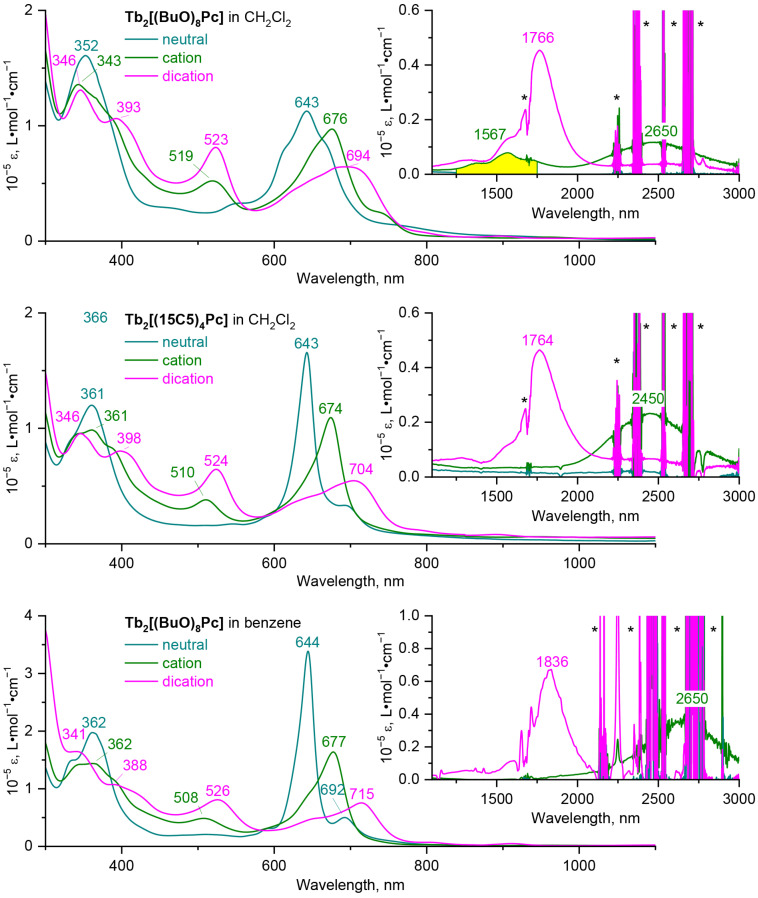
UV-vis-NIR spectra of neutral forms of **Tb_2_[(BuO)_8_Pc]_3_** and **Tb_2_[(15C5)_4_Pc]_3_** and their mono- and dicationic forms, obtained by 1*e*- and 2*e*-oxidation with **OxSbCl_6_**. The marker of the gauche conformation of **Tb_2_[(BuO)_8_Pc]_3_^+^** in CH_2_Cl_2_ is highlighted with yellow colour. Asterisks mark the absorption of solvents.

**Figure 4 molecules-27-06498-f004:**
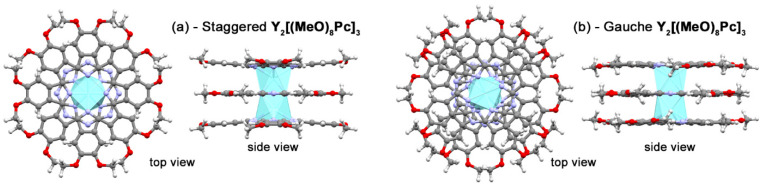
Top and side views on geometries of staggered (**a**) and gauche (**b**) conformations of **Y_2_[(MeO)_8_Pc]_3_** according to BP86/def2-SVP calculations.

**Figure 5 molecules-27-06498-f005:**
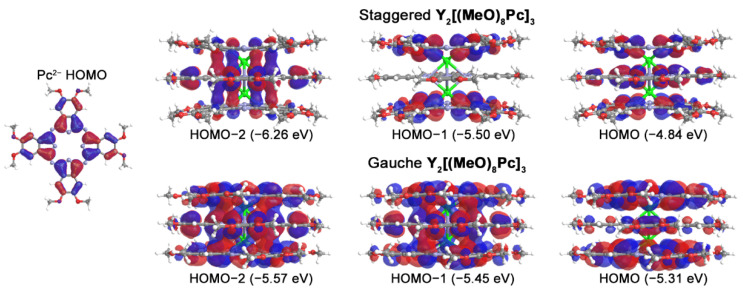
Appearances of highest occupied molecular orbitals of parent Pc^2−^ ligand and neutral **Y_2_[(MeO)_8_Pc]_3_** according to single-point CAM-B3LYP/def2-SVP calculations for geometries optimized at BP86/def2-SVP level of theory.

**Figure 6 molecules-27-06498-f006:**
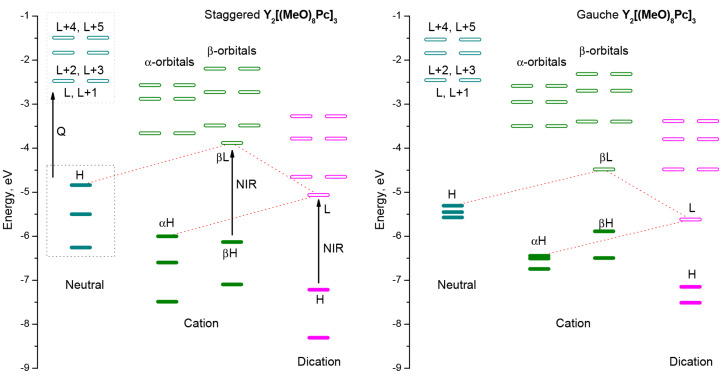
Energies of highest occupied molecular orbitals and redox forms of **Y_2_[(MeO)_8_Pc]_3_** according to single-point CAM-B3LYP/def2-SVP calculations for geometries optimized at BP86/def2-SVP level of theory. H stands for HOMO, L stands for LUMO.

**Figure 7 molecules-27-06498-f007:**
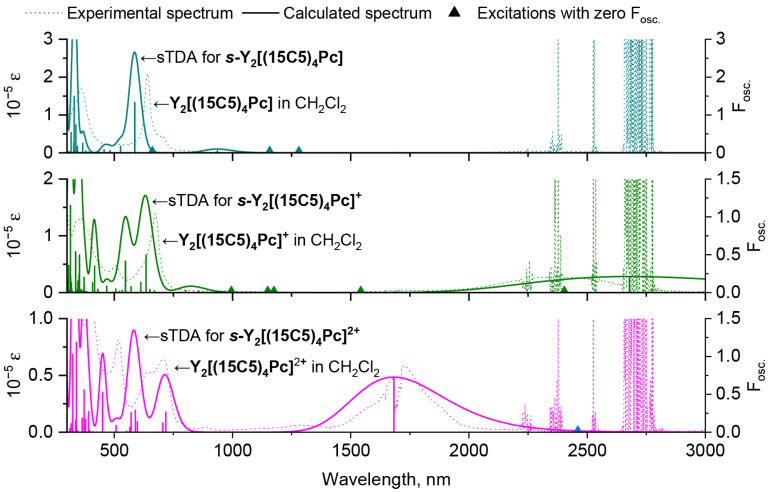
Comparison of experimental spectra of neutral and oxidized **Y_2_[(15C5)_4_Pc]_3_** in CH_2_Cl_2_ and calculated UV-vis-NIR spectra of ***s***-**Y_2_[(MeO)_8_Pc]_3_** according to single-point sTDA CAM-B3LYP/def2-SVP calculations for staggered conformations of respective redox-forms optimized at BP86/def2-SVP level of theory. The shapes of calculated spectra are given with 0.1 eV half-width.

**Figure 8 molecules-27-06498-f008:**
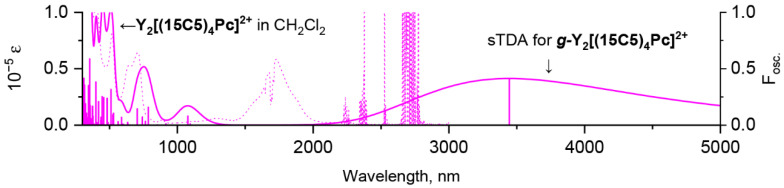
Comparison of experimental spectrum of **Y_2_[(BuO)_8_Pc]_3_^2+^** in CH_2_Cl_2_ and calculated UV-vis-NIR spectrum of ***g***-**Y_2_[(MeO)_8_Pc]_3_^2+^** according to single-point sTDA CAM-B3LYP/def2-SVP calculations for geometry optimized at BP86/def2-SVP level of theory. The shape of calculated spectrum is given with 0.1 eV half-width.

**Figure 9 molecules-27-06498-f009:**
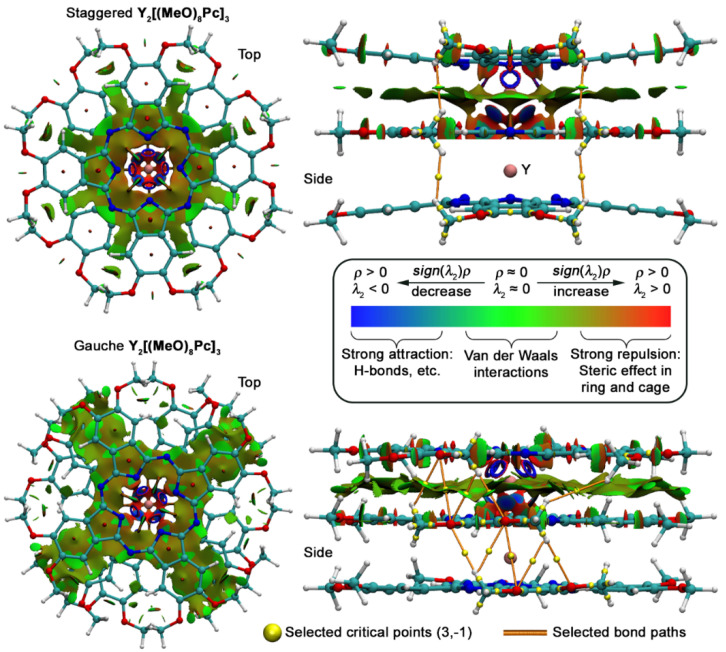
NCI plots for staggered and gauche conformations of **Y_2_[(MeO)_8_Pc]_3_** with selected critical points (3,−1) corresponding to intramolecular interactions of methoxy-groups. The inset shows the qualitative dependence of the isosurface colour on the strength of the noncovalent interaction. For clarity, RDG isosurface is plotted only for the region between two Pc ligands.

**Figure 10 molecules-27-06498-f010:**
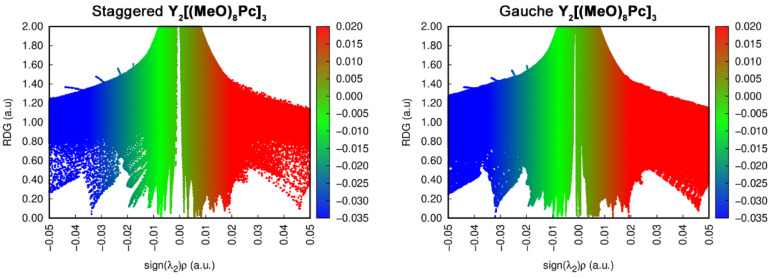
RDG vs. *sign*(*λ*_2_)*ρ* plots for staggered and gauche conformations of **Y_2_[(MeO)_8_Pc]_3_**.

**Table 1 molecules-27-06498-t001:** Difference of energies of gauche and staggered conformers (∆E = E***^g^*** − E***^s^***) of **Y_2_[(MeO)_8_Pc]_3_** according to *r*^2^SCAN-3c single point calculations for geometries optimized at BP86/def2-SVP level of theory. Accounting for solvation was made within SMD for benzene and dichloromethane.

	∆E(gas), kcal/mol	∆E(C_6_H_6_), kcal/mol	∆E(CH_2_Cl_2_), kcal/mol
**Y_2_[(MeO)_8_Pc]_3_**	−13.3	−3.8	−2.5
**Y_2_[(MeO)_8_Pc]_3_^+^**	−5.6	3.2	3.8
**Y_2_[(MeO)_8_Pc]_3_^2+^**	2.4	10.0	10.0

**Table 2 molecules-27-06498-t002:** Selected structural features of redox-forms of **Y_2_[(MeO)_8_Pc]_3_** according to BP86/def2-SVP calculations. The values in brackets correspond to experimental XRD data.

	*d*(Y…Y), Å	*d*(N_4_…N_4_), Å	*θ*, °	*d(*Y…Y), Å	*d*(N_4_…N_4_), Å	*θ*, °
	***s***-Conformer			***g***-Conformer		
**Y_2_[(MeO)_8_Pc]_3_**	3.403 (3.429) ^2^	2.923 (2.952) ^2^	45 ^1^ (43.9) ^2^	3.486 (3.517) ^4^	2.983 (3.028) ^4^	22.6 (33.0) ^4^
**Y_2_[(MeO)_8_Pc]_3_^+^**	3.385	2.898	45^1^	3.459	2.97	23.0
**Y_2_[(MeO)_8_Pc]_3_^2+^**	3.366 (3.435) ^3^	2.873 (2.980)^3^	45 ^1^ (44.2) ^3^	3.442	2.956	22.7

^1^ The angle of 45° comes from geometrical constraints imposed during geometry optimization. ^2^ Data taken from X-ray structure of **[(15C5)_4_Pc]Y[(15C5)_4_Pc]Y(Pc)**, N_4_…N_4_ and *θ* values are averaged (CCRC ITUJEP, [15]). ^3^ Data taken from X-ray structure of **Tb_2_[(BuO)_8_Pc]_3_^2+^(SbCl_6_^−^)_2_** (CCDC FURGUX, [14]). ^4^ Data taken from X-ray structure of **Tb_2_[(BuO)_8_Pc]_3_** (CCDC CAMXUL, [26]).

## Data Availability

Data is contained within the article or Appendix A.

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
