# Peer review of "Redox-Triggered Switching of Conformational State in Triple-Decker Lanthanide Phthalocyaninates"

_molecules, 2022, doi:10.3390/molecules27196498_

Round 1

Reviewer 1 Report

The proposed manuscript of A.G. Martynov et. al. deals with structure and electronic excitation of the triple-decker complexes of diterbium(III). Two different ligands are used to prepare the complexes. Although the prepared complexes were already described in the literature, authors improved the synthesis by adopting different solvents and achieved higher yields and shorter synthesis time. Next, UV-Vis-NIR spectra were measured in two different solvents. The solvent induced changes in spectrum of the Tb2[(BuO)8Pc]3 was interpreted, in accordance with literature, as result of the conformational change (staggered vs. gauche conformation). The complexes were later stepwise oxidized, and the spectral changes were again interpreted. Interpretation is the dominant part of manuscript and is based on the sTD-DFT calculation for a model complex. It is shown how s/g conformation influences absorption spectrum and corresponding excited states are identified.

Article is well written and illustrated. Supporting information provide sufficient data to repeat work of authors.  

Conceptual remarks

1. Number of approximations applied in the analysis is huge, and I am not fully convinced that the possible loss of precision is controlled.

a) Exchange of the central cations Tb([Xe]f9 s2) to Y([Kr]d1 s2) – makes the system different in terms of electronic structure. In the manuscript work of Horii[14] is cited as support for such an exchange, but the cited work is large cooperation of experimental and theoretical methods and Tb/Y exchange is mentioned there only as side information, the same article also uses Tb/Lu exchange in modelling. Consider supporting this decision by more discussion/or calculations.

b) Next, the experiment/theory UV/Vis/NIR spectra are used to identify s/g conformation. Any vibronic influences on the spectrum are neglected and only vertical excitations are used. This approximation makes sense for large complexes, but it also induces some error, together with level of calculation (sTD-DFT). At the end considering possible error bars is the difference between theoretically predicted spectra of s/g significant to prove which one is observed experimentally(Figs 7/8)? I am not sure, please consider adding discussion/calculations which will clarify this issue.

2. Absorption in NIR spectrum can appear because of low energy electronic excitations as interpreted in proposed work, but also as overtone/combination vibrations. With progressive oxidation some electrons are removed from antibonding orbitals, which may(or may not) allow some vibronic transitions.

3. Molecules has currently impact factor almost 5 and became Q1/Q2 journal according to JCR. With this in mind, I believe that proposed manuscript lacks in novelty and significance. I believe presented work is well performed, but it is not, in my opinion novel/significant, as expected for this kind of journal. I would expect either much deeper, more controlled computational study, or more complementary experimental work.

Other questions

4. Figure 2(left) – the main difference in absorption spectrum in benzene/dichloromethane is significant expansion and loss of structure for peak at 644 nm. Could this relate to barrier for ligand rotation around Tb-Tb axis? Stacking benzene hinders this rotation and allows for clear structure of the spectrum while CH2CH2 without specific interactions allows more rotation?

5. Authors contributed with their previous works significantly to the field. Still, I find number of self-citations quite high. Please consider adding works of other authors as well. For example, second part of the introduction, starting from line 60.

6. Line 251-253 – HOMO is considered antibonding with respect to overlap between ligands? The inter-ligand distance variation is very subtle (Table 2) and depends on i) mutual dispersion interactions between ligands, ii) their electrostatic repulsion – as pointed by authors, and iii) electrostatic attraction between Tb(3+) and ligands. When removing electron one decreases ii) but also changes i) and also decreases iii).

Author Response

We would like to thank Editor and Referees for attention paid to our manuscript and helpful comments! All corrections are highlighted, new pdf file with revised version of manuscript is attached.

Sincerely,

Yulia Gorbunova on behalf of authors

Comments and Suggestions for Authors

The proposed manuscript of A.G. Martynov et. al. deals with structure and electronic excitation of the triple-decker complexes of diterbium(III). Two different ligands are used to prepare the complexes. Although the prepared complexes were already described in the literature, authors improved the synthesis by adopting different solvents and achieved higher yields and shorter synthesis time. Next, UV-Vis-NIR spectra were measured in two different solvents. The solvent induced changes in spectrum of the Tb2[(BuO)8Pc]3 was interpreted, in accordance with literature, as result of the conformational change (staggered vs. gauche conformation). The complexes were later stepwise oxidized, and the spectral changes were again interpreted. Interpretation is the dominant part of manuscript and is based on the sTD-DFT calculation for a model complex. It is shown how s/g conformation influences absorption spectrum and corresponding excited states are identified.

Article is well written and illustrated. Supporting information provide sufficient data to repeat work of authors. 

We would like to thank the reviewer for attention paid to our work and valuable comments and suggestions. Please, find our point-to-point replies to your questions.

 Conceptual remarks

  1. Number of approximations applied in the analysis is huge, and I am not fully convinced that the possible loss of precision is controlled.

We do have to use certain approximations to simplify our systems for further computations, but we rely on our previous experience and data reported in literature concerning the appropriate models which can be used for the interpretation of experimental results.

  1. a) Exchange of the central cations Tb([Xe]f9 s2) to Y([Kr]d1 s2) – makes the system different in terms of electronic structure. In the manuscript work of Horii [14] is cited as support for such an exchange, but the cited work is large cooperation of experimental and theoretical methods and Tb/Y exchange is mentioned there only as side information, the same article also uses Tb/Lu exchange in modelling. Consider supporting this decision by more discussion/or calculations.

Indeed, Ref. [14] by Horii et al. reports huge amount of excellent experimental data on oxidized sandwich complexes. However, UV-Vis-NIR spectroscopy was used mainly as an instrument to identify redox states of complexes. Quantum-chemical interpretation of spectral data is given only in Supporting information, maybe without explicit explanations of validity of applied approximations, as the main emphasis was made on structural and magnetic studies.

As for Tb/Lu exchange, it was discussed only in the context of comparison of ESR spectra of oxidized complexes with spectra of supramolecular dimers, studied previously by N. Ishikawa. The replacement of terbium with lutetium was not used for DFT calculations.

As for replacement of terbium with yttrium, it is exactly what the authors did to perform their TD-DFT calculations of model sandwich complexes (see Section 4 of Supporting information for Ref. [14]). The reported calculations of UV-Vis-NIR spectra for these yttrium-substituted models are in excellent agreement with experimental spectra observed for real terbium complexes (Page S267, Fig. S318). The authors show that the excitations correspond to electronic transitions between Pc-centred orbitals (Page S205, Section 4.7).

Then the authors show that the experimentally-registered UV-Vis-NIR spectra of terbium and yttrium complexes are almost identical (Figs. S319 and S320), thus the definitive difference in electronic structure of metal centres themselves do not affect the overall spectral properties of these types of complexes. Therefore, we relied on this approach and used it in our work.

The analogy of spectral properties of yttrium and terbium complexes appear from the fact that excitations responsible for the appearance of bands in UV-Vis-NIR spectra of sandwich complexes, involve Pc-centred frontier p-orbitals [Inorg. Chem. 1999, 38, 3173–3181, doi:10.1021/ic981463x]. The energies of these orbitals and, consequently, wavelengths of corresponding bands in spectra are governed by interligand distances, which in turn depend on the size of metal ion, but not on its orbital structure once rare earth element complexes are concerned [J. Phys. Chem. A 2007, 111, 392–400, doi:10.1021/jp066157g]. Thus, experimentally measured spectra of Tb(III) and Y(III) trisphthalocyaninates have almost analogous appearances due to close ionic radii of these elements – 1.040 and 1.019 pm respectively [Acta Cryst. 1976, A32, 751–767, doi:10.1107/S0567739476001551].

To convince you that our conclusions made for terbium complexes are valid, we synthesized the samples of Y2[(BuO)8Pc]3 and Y2[(15C5)4Pc]3 and studied their oxidation. The observed correlations between the oxidation state, solvent nature and spectral appearance are analogous to those reported for terbium counterparts. New spectral data were added both to the manuscript and Supporting information section.

  1. b) Next, the experiment/theory UV/Vis/NIR spectra are used to identify s/g conformation. Any vibronic influences on the spectrum are neglected and only vertical excitations are used. This approximation makes sense for large complexes, but it also induces some error, together with level of calculation (sTD-DFT). At the end considering possible error bars is the difference between theoretically predicted spectra of s/g significant to prove which one is observed experimentally(Figs 7/8)? I am not sure, please consider adding discussion/calculations which will clarify this issue.

Indeed, we perform only vertical excitation calculations, as prediction of vibrationally-resolved TD-DFT spectra is prohibitive from the viewpoint of our computational facilities, moreover simplified TD-DFT (sTD-DFT and its sTDA approximation) simply cannot perform such calculations. Thus, we agree that we cannot fully reproduce the band shapes as we neglect vibrational overtones. To our best knowledge vibrationally-resolved TD-DFT calculations for sandwich complexes have never been reported. However, the aforementioned studies published by groups of Ishikawa, Jiang and Yamashita clearly demonstrate that vertical excitations calculations are sufficient for interpretation of spectra of sandwich complexes.

On the other hand, it is in principle impossible to achieve complete agreement between calculated and experimental spectra, and even 0.2 eV error between calculated and predicted bands are commonly acceptable for full TD-DFT. Moreover, this error can be different for different parts of spectra as was demonstrated by Prof. Victor Nemykin in his recent comprehensive publication [J. Phys. Chem. A 2019, 123, 132–152, doi:10.1021/acs.jpca.8b07647]. However, these calculations do not aim to reproduce the spectral appearance, but they must reproduce the trends in changes observed in experimental UV-Vis-NIR spectra upon structural modifications.

We have previously shown that sTDA and sTD-DFT approximations with CAM-B3LYP functional and double-z basis sets is a very convenient tool to reproduce such trends with errors not exceeding 0.02 eV in some cases [ACS Omega 2019, 4, 7265–7284, doi:10.1021/acsomega.8b03500]. In our work we show that the simplified approximation can be used to interpret spectra of redox-forms of sandwich complexes to identify their conformational states (Fig. 7), the errors in Q-band in NIR regions are in the ranges of 0.08-0.18 eV and 0.03-0.04 eV respectively. In Fig. 8 we show the theoretical spectrum of gauche conformer of triple-decker dication suggesting that this form is experimentally unobtainable at least within the studied approach.

It is also important to emphasize that simplified TD-DFT methods allow for impressive acceleration compared to full TD-DFT for huge molecules, making these calculations more accessible to the general scientific community.

  1. Absorption in NIR spectrum can appear because of low energy electronic excitations as interpreted in proposed work, but also as overtone/combination vibrations. With progressive oxidation some electrons are removed from antibonding orbitals, which may(or may not) allow some vibronic transitions.

See reply to question 1b concerning the vibronic transitions.

  1. Molecules has currently impact factor almost 5 and became Q1/Q2 journal according to JCR. With this in mind, I believe that proposed manuscript lacks in novelty and significance. I believe presented work is well performed, but it is not, in my opinion novel/significant, as expected for this kind of journal. I would expect either much deeper, more controlled computational study, or more complementary experimental work.

To prove the conclusions which we made within our approximations, we add experimental data on yttrium complexes. As for significance, studies of sandwich complexes in general is a hot topic in modern phthalocyanine chemistry due to their high potential in SMM and spintronic area. Although we do not take advantages of magnetism of these complexes in our work, we believe that this work expands the results reported Ref. [14] by Horii et al. in terms of spectroscopy and tuning of properties of sandwich complexes. Ref. 14 shows that certain conformations can be stabilized in crystal phase, but we show that this is true for solutions as well. The theoretical correlations between conformational state of sandwiches and their nonlinear optical properties have been reported [ChemPhysChem 2015, 16, 1889–1897, doi:10.1002/cphc.201500082.], but their experimental validity has not been verified yet because of the lack of tools affording dynamic control of sandwich complexes conformations in solution. We believe that these factors justify the novelty and significance of our work.

Other questions

  1. Figure 2(left) – the main difference in absorption spectrum in benzene/dichloromethane is significant expansion and loss of structure for peak at 644 nm. Could this relate to barrier for ligand rotation around Tb-Tb axis? Stacking benzene hinders this rotation and allows for clear structure of the spectrum while CH2CH2 without specific interactions allows more rotation?

Yes, the rotation of ligands around M-M axes of sandwich complexes is likely to occur in solution, but the distribution of conformers is shifted either to s- or g- depending on the interactions with solvent molecules. We added corresponding comments to the manuscript.

  1. Authors contributed with their previous works significantly to the field. Still, I find number of self-citations quite high. Please consider adding works of other authors as well. For example, second part of the introduction, starting from line 60.

We added more references concerning the spectroscopy of siloxane-based bisphthalocyaninates (group of Prof. Nagao Kobayashi) and crown-substituted phthalocyanines (group of Prof. Jianzhuang Jiang).

  1. Line 251-253 – HOMO is considered antibonding with respect to overlap between ligands? The inter-ligand distance variation is very subtle (Table 2) and depends on i) mutual dispersion interactions between ligands, ii) their electrostatic repulsion – as pointed by authors, and iii) electrostatic attraction between Tb(3+) and ligands. When removing electron one decreases ii) but also changes i) and also decreases iii).

Yes, we differentiate the bonding, nonbonding or antibonding nature of orbitals with respect to their overlap. For sure, all three factors affect the orbital diagrams, but the main factor here is molecular symmetry – it leaves behind other subtle structural effects.

Reviewer 2 Report

This manuscript reports the study of multi-decker complexes of Tb exploring their spectroscopic properties mostly by measuring the corresponding UV-Vis-NIR spectra. From this point of view the work seems adequately designed and the research properly addressed. Authors also assume that structural parameters such as interligand distance and skew angle.

For the DFT calculations, authors relied on a model system using Y(III), which is the main issue I find with this manuscript. 

Authors support the choice of using Y(III) due to the simplicity of handling the "ambiguity of the spin states". And here resides the issue I found most critical with this work. So, authors discuss their data in terms of magnetism and how it affects the observable properties. Next, they neglect it based on the simple fact that for simplicity they replace a paramagnetic entity with another that is not to explain something that is born from that property. Furthermore, the orbitals (4f) in Tb(III) are completely different from those in Y(III) and this is also critical in the coordination, which will be reflected in the interligand distances and skew angles. 

I could understand that Y(III) could be considered a RE metal (as it is found in Nature with proper RE ores, but it is not in terms of electronic or magnetic properties and therefore this is a critical issue with this work.

What authors need to do this test the spin states and then run their study. As it is at present, I find it baseless.

Furthermore, in the oxidation study a description of the experimental protocol is missing and must be added. In addition, authors mention oxidation towards cation and di-cation but how is this controlled experimentally, i.e., how are authors sure that in both cases there is not contamination of the cation by some di-cation or in di-cation all cation was properly oxidized? This must be explained as well.

Therefore, I recommend rejection of this manuscript allowing authors to improve their work

Author Response

We would like to thank Editor and Referees for attention paid to our manuscript and helpful comments! All corrections are highlighted, new pdf file with revised version of manuscript is attached.

Sincerely,

Yulia Gorbunova on behalf of authors

Comments and Suggestions for Authors

This manuscript reports the study of multi-decker complexes of Tb exploring their spectroscopic properties mostly by measuring the corresponding UV-Vis-NIR spectra. From this point of view, the work seems adequately designed and the research properly addressed. Authors also assume that structural parameters such as interligand distance and skew angle.

For the DFT calculations, authors relied on a model system using Y(III), which is the main issue I find with this manuscript. 

Authors support the choice of using Y(III) due to the simplicity of handling the "ambiguity of the spin states". And here resides the issue I found most critical with this work. So, authors discuss their data in terms of magnetism and how it affects the observable properties. Next, they neglect it based on the simple fact that for simplicity they replace a paramagnetic entity with another that is not to explain something that is born from that property. Furthermore, the orbitals (4f) in Tb(III) are completely different from those in Y(III) and this is also critical in the coordination, which will be reflected in the interligand distances and skew angles. 

I could understand that Y(III) could be considered a RE metal (as it is found in Nature with proper RE ores, but it is not in terms of electronic or magnetic properties and therefore this is a critical issue with this work.

What authors need to do this test the spin states and then run their study. As it is at present, I find it baseless.

Therefore, I recommend rejection of this manuscript allowing authors to improve their work

We would like to thank you for attention paid to our work, although you have negative impression on our manuscript. We would like to make some comments to convince you in validity of our approximations.

Although we do start the manuscript with correlations between structural and magnetic properties of terbium phthalocyaninates, we do not take the advantages of magnetic behavior of these complexes. We demonstrate how UV-Vis-NIR spectroscopy in solution can be used to identify the conformational states of complexes.

To prove that UV-Vis-NIR spectra of sandwich trisphthalocyaninates do not depend on the nature of the metal center (Tb([Xe]f9s2 or Y([Kr]d1s2) we synthesized the samples of Y2[(BuO)8Pc]3 and Y2[(15C5)4Pc]3 and studied their oxidation. The observed correlations between the oxidation state, solvent nature and spectral appearance are analogous to those reported for terbium counterparts. New spectral data were added to Supporting information section.

The analogy of spectral properties of yttrium and terbium complexes appear from the fact that excitations responsible for the appearance of bands in UV-Vis-NIR spectra of sandwich complexes, involve Pc-centred frontier p-orbitals [Inorg. Chem. 1999, 38, 3173–3181, doi:10.1021/ic981463x]. The energies of these orbitals and, consequently, wavelengths of corresponding bands in spectra are governed by interligand distances, which in turn depend on the size of metal ion, but not on its orbital structure once rare earth element complexes are concerned [J. Phys. Chem. A 2007, 111, 392–400, doi:10.1021/jp066157g]. Thus, experimentally measured spectra of Tb(III) and Y(III) trisphthalocyaninates have almost analogous appearances due to close ionic radii of these elements – 1.040 and 1.019 pm respectively [Acta Cryst. 1976, A32, 751–767, doi:10.1107/S0567739476001551].

Analogous approach with Tb→Y replacement was used in Ref. [14] by Horii et al [Chem. – A Eur. J. 2020, 26, 8621–8630, doi:10.1002/chem.202001365]. The authors did it to perform TD-DFT calculations of model triple, quadruple and quintuple-decker complexes (see Section 4 of Supporting information for Ref. [14]). The reported calculations of UV-Vis-NIR spectra for these yttrium-substituted models are in excellent agreement with experimental spectra observed for real terbium complexes (Page S267, Fig. S318). The authors show that the excitations correspond to electronic transitions between Pc-centred orbitals (Page S205, Section 4.7).

Thus, calculations of UV-Vis-NIR spectra can be reliably performed for diamagnetic analogues of paramagnetic terbium complexes, however we do understand that such replacement will no longer be applicable for calculations of magnetic properties which is out of the scope of our work at the current stage.

Furthermore, in the oxidation study a description of the experimental protocol is missing and must be added. In addition, authors mention oxidation towards cation and di-cation but how is this controlled experimentally, i.e., how are authors sure that in both cases there is not contamination of the cation by some di-cation or in di-cation all cation was properly oxidized? This must be explained as well.

We added description of experimental protocol used for oxidation. The coexistence of only two forms neutral/cationic or cationic-dicationic at different stages of spectrophotometric titrations was confirmed by observation of individual sets of isosbestic points at these stages. Moreover, our spectral data are in agreement with those obtained in Ref. [14] by electrochemical oxidation at certain potentials.

Reviewer 3 Report

The manuscript describes the study of lanthanide phthalocyaninates using spectroscopic and computational experiments. The paper provides a detailed introduction to the study, which clearly reflects the available literature data and the objectives of the work. Despite the study of already known compounds (importantly, their syntheses were well optimized in this work), the authors develop the chemistry of such complexes in order to expand their practical potential. The data obtained undoubtedly demonstrate the practical interest of combining different methods and techniques for interpretation and establishment of spectroscopies results indicating different conformational states, which are of great important for physical-chemical properties of lanthanide phthalocyaninates.

Minor comments:

1)       Line 19 and 22 change “allowed to” to “allowed us to”.

2)      Perhaps it would be more informative to present the data in Figure 2 not as a normalized extinction, but through the extinction coefficient.

3)      Figure 9 and 10 change the formula of compound from “Y2[MeO)8Pc]3” to “Y2[(MeO)8Pc]3”.

I believe this work can be published in the journal Molecules after minor revision.

Author Response

We would like to thank Editor and Referees for attention paid to our manuscript and helpful comments! All corrections are highlighted, new pdf file with revised version of manuscript is attached.

Sincerely,

Yulia Gorbunova on behalf of authors

Comments and Suggestions for Authors

The manuscript describes the study of lanthanide phthalocyaninates using spectroscopic and computational experiments. The paper provides a detailed introduction to the study, which clearly reflects the available literature data and the objectives of the work. Despite the study of already known compounds (importantly, their syntheses were well optimized in this work), the authors develop the chemistry of such complexes in order to expand their practical potential. The data obtained undoubtedly demonstrate the practical interest of combining different methods and techniques for interpretation and establishment of spectroscopies results indicating different conformational states, which are of great important for physical-chemical properties of lanthanide phthalocyaninates.

Minor comments:

1)       Line 19 and 22 change “allowed to” to “allowed us to”.

Corrected

2)      Perhaps it would be more informative to present the data in Figure 2 not as a normalized extinction, but through the extinction coefficient.

Plotting spectra in normalized optical densities makes it easier to see the difference in spectral profiles. Thus, we prefer to keep these spectra the way they are already presented, but we measured extinction coefficients for all complexes and added them to experimental section.

3)      Figure 9 and 10 change the formula of compound from “Y2[MeO)8Pc]3” to “Y2[(MeO)8Pc]3”.

Thank you for noting the error, pictures were corrected.

I believe this work can be published in the journal Molecules after minor revision.

Thank you for positive evaluation of our work!

Round 2

Reviewer 1 Report

I would like to thank the authors for providing well-grounded answers, performing additional experiments, and improving the discussion. Now I support their choice of computational model and interpretation of the results. I believe that the manuscript fulfils the criteria for accepting in Molecules.

Reviewer 2 Report

Authors have adequately provided enough and solid justification for the questions raised in the first revision round. They have clarified less straightforward options in alongside the design and the discussion of the research, by providing solid insights to support their options.

Therefore, the manuscript can be accepted for publication.